

# Nanopublication-based semantic publishing and reviewing: a field study with formalization papers

Cristina-Iulia Bucur[1], Tobias Kuhn[1], Davide Ceolin[2] and Jacco van Ossenbruggen[1]

[1] Computer Science Department, Vrije Universiteit Amsterdam, Amsterdam, The Netherlands
[2] Human-Centered Data Analytics Group, Centrum Wiskunde & Informatica, Amsterdam, The Netherlands

## ABSTRACT

With the rapidly increasing amount of scientific literature, it is getting continuously more difficult for researchers in different disciplines to keep up-to-date with the recent findings in their field of study. Processing scientific articles in an automated fashion has been proposed as a solution to this problem, but the accuracy of such processing remains very poor for extraction tasks beyond the most basic ones (like locating and identifying entities and simple classification based on predefined categories). Few approaches have tried to change how we publish scientific results in the first place, such as by making articles machine-interpretable by expressing them with formal semantics from the start. In the work presented here, we propose a first step in this direction by setting out to demonstrate that we can formally publish high-level scientific claims in formal logic, and publish the results in a special issue of an existing journal. We use the concept and technology of nanopublications for this endeavor, and represent not just the submissions and final papers in this RDF-based format, but also the whole process in between, including reviews, responses, and decisions. We do this by performing a field study with what we call formalization papers, which contribute a novel formalization of a previously published claim. We received 15 submissions from 18 authors, who then went through the whole publication process leading to the publication of their contributions in the special issue. Our evaluation shows the technical and practical feasibility of our approach. The participating authors mostly showed high levels of interest and confidence, and mostly experienced the process as not very difficult, despite the technical nature of the current user interfaces. We believe that these results indicate that it is possible to publish scientific results from different fields with machine-interpretable semantics from the start, which in turn opens countless possibilities to radically improve in the future the effectiveness and efficiency of the scientific endeavor as a whole.

## INTRODUCTION

Considering the abundance of scientific articles that are published every day (*Uddin, Khan & Baur, 2015*), keeping up with the latest research is becoming a significant challenge for researchers in many fields. This is at least partially due to the fact that we are still holding on to an archaic paradigm of scientific publishing: the canonical way to publish scientific results

Corresponding author
Cristina-Iulia Bucur, c.i.bucur@vu.nl

is by writing them up in long English texts called articles, which are in the best case easy to read by human experts but remain mostly inaccessible to automated approaches (except on a very superficial level with text mining approaches) (*Bhargava et al., 2017*; *Westergaard et al., 2018*; *Shukkoor, Raja & Baharuldin, 2022*). These articles then undergo peer reviewing, which is typically done in a way that is secretive and not standardized, with the effect that the reviewing process may lack transparency and that valuable comments from the reviewers cannot be reused or built upon. There have been studies on the effectiveness of peer-reviewing in its current form (*Smith, 1988*; *Linkov, Lovalekar & LaPorte, 2006*; *Kotturi et al., 2017*) that showed not only systematic biases among peer-reviewers, but also a lack of transparency in the general peer-reviewing process as a whole (*Smith, 2010*; *Benda & Engels, 2011*; *Lee et al., 2012*). Making reviews open might alleviate some of these concerns by ensuring higher-quality reviews, while at the same time increasing the trust in the reviewing process and the quality of the scientific publications themselves.

A range of approaches have been proposed to address some of these problems by making scientific texts machine-readable, allowing for automatic summarising, finding and retrieving information easier and even the ability to (partially) reason on the scientific texts themselves. Text mining approaches work reasonably well when it comes to simple entity extraction with techniques like named-entity recognition to extract the main concepts from a text (*e.g.*, *Al-Moslmi et al., 2020*; *Yadav & Bethard, 2018*), but accuracy dramatically drops with more complicated tasks like relation extraction or identifying links between entities (*Etzioni et al., 2005*; *Xu et al., 2015*; *Zeng et al., 2014*).

The vast majority of existing approaches of making scientific texts machine-readable have one thing in common: they take the current paradigm of scientific articles for granted and therefore take them as their starting point to extract information. While it is important to try to process the vast amount of existing scientific literature that has the form of long English texts (and sometimes long texts in other languages), we should also think about how we can improve the way how we publish scientific insights in the first place. An important aspect of this is the vision of semantic publishing, which we mean here in the sense of *genuine semantic publishing* (*Kuhn & Dumontier, 2017*), where the machine-interpretable formal semantics cover the main scientific claims the work is making. Nanopublications (*Groth, Gibson & Velterop, 2010*), which are small RDF-based semantic packages, have emerged as a powerful concept and technology for enabling such genuine semantic publishing. We should note here that we are using the term "semantic" in its more narrow sense of "with meaning represented in a formal computer-interpretable manner", and not in the more general meaning of "with respect to meaning".

In previous research we have applied nanopublications to implement a semantic and fine-grained model for reviewing (*Bucur, Kuhn & Ceolin, 2019*), and have extended this to semantically represent the full structure of (classical) scientific articles with their reviews and review responses as a single network of nanopublications (*Bucur et al., 2020*). In order to get closer to our vision of genuine semantic publishing, however, we need to represent not just the structure but also the main content of these articles, most importantly their main scientific claims. To that aim, we proposed in subsequent work the *super-pattern*, a

semantic template to represent the meaning of scientific claims in formal logic (*Bucur et al., 2021*).

Taking an example from our previous study as an illustration of the super-pattern, it has been stated in scientific literature (*Felix & Barrand, 2002*) that in particular kinds of cells in the rat brain (specifically, endothelial cells) some sort of stress called transient oxidative stress affects the expression of a protein called Pgp. The super-pattern consists of five slots that would in this example be filled in as follows:

- Context class: rat brain endothelial cell
- Subject class: transient oxidative stress
- Qualifier: generally
- Relation: affects
- Object class: Pgp expression

Informally, we can read this in the following way: whenever there is an instance of transient oxidative stress in the context of an instance of a rat brain endothelial cell, then generally (meaning in at least 90% of the cases), that instance of stress has the relation of affecting an instance of Pgp expression. Formally, it directly maps to this logic formula:

$$P( \exists z(\text{pgp-expression}(z) \wedge \text{in-context}(z,x) \wedge \text{affects}(y,z)) \mid$$

$$\text{transient-oxidative-stress}(y) \wedge \text{rat-brain-endothelial-cell}(x) \wedge \text{in-context}(y,x) ) \geq 0.9.$$

This is stating in logic terms (in slightly non-standard notation using conditional probability as a shorthand) that given a thing $y$ of type *transient-oxidative-stress* in the context of a thing $x$ of type *rat-brain-endothelial-cell*, the probability of there being a $z$ of type *pgp-expression* that is in the same context $x$ is at least 90%. We have shown that this pattern can be applied to formalize most high-level claims found in scientific literature across disciplines (*Bucur et al., 2021*).

The representation above is in a certain way more precise than what the article was stating, by making the percentage of 90% explicit. For this example, we made a best guess, but ideally this decision should of course come from the original researchers. In another sense, the statement is probably less specific than what can be read in the paper in certain other respects, as any formal representation trades to a certain extent nuance and detail for precision. As a further side remark, it is important to realize that the above number of 90% is part of what the statement expresses (namely the minimum ratio of how often the *affects*-relation holds in the given condition), and does *not* stand for the certainty or validity of the statement as a whole. Expressing the (un)certainty of the statement can be done with models such as ORCA (*de Waard & Schneider, 2012*), but this is outside of the scope for the super-pattern.

In the work to be presented below, the main research goal is to combine all the elements we have previously worked on—namely semantic representation of reviews (*Bucur, Kuhn & Ceolin, 2019*), scientific works as networks of nanopublications (*Bucur et al., 2020*), and representing the main claims with the super-pattern (*Bucur et al., 2021*)—in order to implement genuine semantic publishing and putting it to the test in a field study. Whereas

our current scientific publishing process works with narrative-based, natural language texts in the form of scientific articles that are later made more machine-interpretable by means of semantic annotations and semantic interlinking to enhance their semantic integration and discovery, we propose a different approach, one that considers semantics from the beginning. Therefore, the main aim of this research is to make a first step in the direction of publishing with formal semantics from the start, showing that it is possible to represent *semantically* not only scientific claims, but also the entire scientific publishing process without going through other intermediary semantic processing stages. For practical reasons, we did not require the scientific claims in this field study to be novel ones, but they were selected from existing publications. This field study led to the publication of a special issue in an established journal (Data Science) at an established publisher (IOS Press). This special issue consists of what we call *formalization papers*, which are nanopublication-based semantic publications whose novelty lies in the formalization of a previously published scientific claim.

In this research we therefore aim to answer the following research question:

- Can nanopublications and the super-pattern enable a new paradigm of scientific communication where authors publish their scientific findings with formal semantics from the start?

The rest of this article is structured as follows. In 'Background' we describe the current state of the art in the field of scientific publishing with regard to scientific knowledge representation, semantic publishing and semantic articles and also alternative proposed machine-readable approaches like nanopublications. In 'Methodology' we describe our approach with regard to a new way of publishing, starting from a formal way of representing the content of scientific claims and ending with the representation of the publication process itself in what we call "formalization papers". We then report and discuss the results of the field study we performed using formalization papers in 'Results'. Future work and conclusion of the present research are outlined in 'Discussion and Conclusion'.

## BACKGROUND

We provide here the background on scientific knowledge representation, scientific publishing, nanopublications, and genuine semantic publishing in particular.

### Scientific knowledge representation

Novel proposals for the current "Disruption Era" (*Rahardja, Lutfiani & Juniar, 2019*) include scientific publication management models that connect abstract knowledge with actual world problems in the constantly growing body of scientific knowledge (*Chi et al., 2018*), and the use of decentralized publication systems for open science using, for example, existing technologies like Blockchain and IPFS (*Tenorio-Fornés et al., 2019*).

A range of methods have been proposed to make scientific articles more machine-readable: from structuring scientific works as Research Objects (RO) (*Bechhofer et al., 2013*; *Belhajjame et al., 2015*) to using facets in order to uncover the main methods, data, code and other objects that are used in scientific articles (*Peroni et al., 2013*; *McGregor,*

*2008*). Most approaches, however, have focused on automated content extraction from scientific articles as they are currently available. Recent machine learning techniques, for example, can after training with large sets of scientific articles extract the main concepts and structure of scientific articles (*Xu et al., 2015*; *Zeng et al., 2014*). While the results can be very valuable there are also clear limitations, with the resulting data needing almost always manual curation to achieve decent quality (*Garcia-Castro et al., 2013*; *Coulet et al., 2011*; *Sernadela et al., 2015*).

A significant number of vocabularies and ontologies in many various domains have been developed, which are now ready to be used for scientific knowledge representation. But, even though practical problems like finding, accessing and versioning among other things have been reported (*Garijo & Poveda-Villalón, 2020*; *Halpin et al., 2010*; *Hitzler & van Harmelen, 2010*; *Jain et al., 2010*), these vocabularies and ontologies have proven to be extremely useful for example for biomedical literature curation (*Slater, 2014*; *Müller et al., 2018*). A considerable amount of attention has also been given to the datasets accompanying scientific articles. The Data Set Knowledge Graph (DSKG), for example, covers datasets from over 600k scientific publications (*Färber & Lamprecht, 2021*). An important development in this respect is the strong momentum behind the FAIR initiative to make research data findable, accessible, interoperable, and reusable (*Wilkinson et al., 2016*). A large amount of research is ongoing on how these FAIR principles can be put into practices (*e.g.*, *Garijo & Poveda-Villalón, 2020*). Many other aspects of scientific communication have been approached with more formal representations, such as declaring authorship contributions with the contributor roles taxonomy (*McNutt et al., 2018*) to mention just one of them.

Semantic technologies have been used extensively in the Life Sciences, *e.g.*, for the representation and discovery of concepts, their relations and associated supporting evidence in order to integrate distributed repositories (*Hannestad et al., 2021*). A variety of controlled vocabularies exist in these fields that can serve as the foundation to represent scientific knowledge in a structured way in order to semantically capture the context of scientific findings (*Chibucos et al., 2014*; *Slater & Song, 2012*; *Madan et al., 2019*).

The BEL language (*Slater, 2014*) is one of the few attempts to represent the high-level scientific claims themselves, with coverage for specific kinds of biological relations. One of the first attempts that follows the "genuine semantic publishing" vision with a focus on scientific findings from the life sciencies field is the Biological Expression Language (BEL) (*Slater, 2014*). BEL is a language that was developed to express in a computable format scientific findings, being used initially mainly for curation purposes of biological data and later for more complex tasks. Recent research has shifted the initial curation purpose of BEL into multiple development directions, showing the full potential that such computable representations can entail, despite being still limited to the life sciencies field: from software and visualization (*Hoyt, Domingo-Fernández & Hofmann-Apitius, 2018*), to algorithms and analytical frameworks (*Zucker et al., 2021*), data integration (*Domingo-Fernández et al., 2018*), natural language processing (*Shao et al., 2021*), curation workflows (*Hoyt et al., 2019*), to content and applications (*Khatami et al., 2021*).

Also many other domains besides the Life Sciences have adopted the principles and technologies of Linked Data and the Semantic Web, for example to build interlinked, heterogeneous, and semantically rich datasets in Cultural Heritage (*Hyvönen, 2012*) and to find, address, and sometimes even solve research problems in Digital Humanities in interactive ways (*Hyvönen, 2020*).

## Semantic publishing and semantic papers

Semantic publishing applies semantic technology to scientific publishing, and comes in many forms and does not always align with what we have introduced above as genuine semantic publishing (*Kuhn & Dumontier, 2017*). Under this umbrella of semantic publishing, there are approaches that generate semantically-enriched data models from digital publications for the integration, sharing, management and data comparison between publications (*Perez-Arriaga, 2018*), study the semantic annotation and enhancement of scholarly articles (*Shotton, 2009*), provide dynamic visualizations in semantically enhanced papers (*Senderov & Penev, 2016*), assess the versioning aspect of semantic publishing (*Papakonstantinou, Fundulaki & Flouris, 2018*), create a global-scale platform with a dataset metadata for automated ingestion, discover, and linkage (*Jacob, Griffith & Le, 2017*), and propose semantic and web-friendly HTML-based alternatives to the currently PDF-focussed scientific writing process (*Peroni et al., 2016*). Semantic enhancements of scientific articles can be used for semantic interlinking, interactive figures, re-orderable references and even summary creation (*Shotton et al., 2009*), and workflows to convert regular scientific articles into linked open data have also been investigated (*Sateli & Witte, 2016*). Other approaches like the compositional and iterative semantic enhancement (CISE) advocate for a process of automatic semantic enhancement with semantic annotations (*Peroni, 2017*). A key role in most of these approaches is played by the variety of existing ontologies covering many different aspects of scientific publishing, most importantly the Semantic Publishing and Referencing (SPAR) Ontologies (*Peroni, 2014*).

Further note-worthy approaches include the work to semantically represent the setup and results of scientific studies, which then allows for running meta-analyses in a semi-automated way, better research replication, and automated hypothesis generation (*Tiddi, Balliet & ten Teije, 2020*), and the development of the Open Research Knowledge Graph (*Jaradeh et al., 2019*). The latter is an initiative that aims to make research articles machine-readable by expressing their main scientific entities as a semantically interconnected knowledge graph. This graph is populated by methods such as extracting scientific concepts from the abstracts of scientific articles with the help of annotators (*Brack et al., 2020*).

## Nanopublications

Nanopublications (*Groth, Gibson & Velterop, 2010*) are a specific concept and technology that deserves special attention here. They have been proposed to express scientific (and other kind of) knowledge in Linked Data as small independent publication packages. They allow for rich provenance and metadata and are structured as follows: the assertion part contains the main content of the nanopublication, such as a scientific claim, expressed as

RDF triples. The provenance part of a nanopublication describes how the assertion came about, *e.g.*, by linking to the scientific methods used to arrive at the finding. The publication information part, finally, contains metadata about the nanopublication as a whole, such as by whom and when it was created. Nanopublications can be used for scientific findings, but also for representing the other elements of the scientific workflow, such as reviews and method descriptions, and more generally any kind of small coherent set of RDF triples (*Kuhn et al., 2013*). It has been shown how nanopublications can be made reliable and immutable by identifying them with cryptographic Trusty URIs (*Kuhn & Dumontier, 2015*), and how this allows for a decentralized network of services and template-based user interfaces such as Nanobench (*Kuhn et al., 2021*).

## Genuine semantic publishing

In the genuine semantic publishing vision (*Kuhn & Dumontier, 2017*) semantics are taken into account before, during and after publication and these pertain integrally to the publication itself. As such, genuine semantic publishing means not only to formally represent from the start the structure and content of authentic fine-grained representations of research work, but also to publish these semantic representations (directly by the authors themselves) as main elements of a published entity without the need for a separate narrative article to exist. For most current approaches, in contrast, semantics are considered only after the publication of scientific articles, with semantic annotations, semantic interlinking and semantic integration used to semantically enrich and extract information from research that is still being published in coarse-grain texts in natural language.

One of the first attempts that aligns with some of the elements of genuine semantic publishing is the markup-language TaxPub (*Penev et al., 2012*). TaxPub enables the publication of specific structured information for biological systems, but this is meant only as a tool that is able to provide semantic enhancements for already published scientific articles, hence not the publication of formal fine-grained authentic work by itself. More recently, a set of more advanced and complex tools for biodiversity literature have been created, like the Open Biodiversity Knowledge Management System (OBKMS) (*Penev et al., 2018*). This system is able to not only link reusable data with its provenance, but also to provide a platform for the complete publishing process of biodiversity data from its submission to its reviewing, to its publication and dissemination. In this way, the complete publication process of scientific findings where facts and claims are linked to their original publications is supported by using a community accepted interoperable open format for biodiversity data. Furthermore, OpenBiodiv (*Penev et al., 2019*), an OBKMS that uses a linked open dataset generated from scientific literature is able to provide an infrastructure where biodiversity knowledge can be managed, but this is also based on data extracted from already published scientific articles.

Another project that comes quite close to the genuine semantic publishing vision, even though it does not comply with it completely, is the creation of the Science Knowledge Graph Ontologies (SKGO) (*Fathalla, Auer & Lange, 2020*), an initiative that seeks to organize the scholarly information published online in terms of its content, with a focus on the representation of scientific findings from various fields. As such, workflows that

use semi-automatic methods to capture the contents of research findings in a structured manner have been proposed (*Vahdati et al., 2019*), but again, their core assumption is that this knowledge needs to have been previously published in research articles.

To conclude, apart from very few exceptions such as the work done in the biodiversity data field and the SKGO project, hence on restricted fields and restricted kinds of claims, genuine semantic publishing remains a vision for which we have little practical evidence as of now on how it can work in practice. So, there is still a huge gap with regard to making scientific knowledge machine interpretable, despite all these useful attempts and approaches that aim towards machine interpretability. As such, for a publication to be genuinely semantic, not only the semantic representation from the start of the structure and content of a fine-grained and authentic primary component of a publication entity needs to be considered, but also all the aspects that pertain to its publication. And, as we noticed in the projects mentioned above, the combination of all these requirements together is not present in current research.

# METHODOLOGY

In this section we describe the approach and methods we followed to investigate whether nanopublications and the super-pattern are suitable to achieve genuine semantic publishing.

## Approach

In our approach, we committed to a number of features. First, we wanted the final contributions to be published as "real" papers in a real established journal. They should be fully semantically represented (in RDF) but also have classical views that makes them look like other papers. Like that, they should also seamlessly integrate with the existing bibliometric system and it should be straightforward to cite them in the classical way.

Second, we decided to fully focus on arguably the most interesting element of scientific articles, which happens to also be one of the most challenging to formally represent: the main scientific claims the article is making. Scientific articles have a large number of other interesting pieces of information, *e.g.*, information about the used methods among many other things, but for the purpose of the study to be presented, we focus only on the main claims.

Third, in order to retain the flexibility and power of nanopublications, we decided to refrain from providing a custom-built and optimized user interface that hides the complexity and limits the flexibility. By using generic template-based nanopublication tools and by customizing them solely by providing the templates, we hoped to get a better understanding of how the nanopublication technology works for such kinds of content and workflows in general, and not just for our specific case. On the other hand, this also means that we were looking for a bit more technically minded authors who can handle interfaces that do not come with all the comfort of polished specific applications.

Fourth, we wanted to test a system that *could* be used to publish novel claims, but decided for practical reasons to focus on formalizing claims from previously published articles.

Our approach is therefore based on what we call *formalization papers* that contribute novel formalizations of existing claims.

Finally, we wanted to cover not just these main claims, but the whole publishing workflow that involves the initial submission of contributions, their reviewing, the responses to the reviews, the updated versions, and the final decision, and represent these as independent but interlinked nanopublications.

## Formalization papers

Our approach builds upon our new concept of formalization papers. A formalization paper contributes a semantic formalization of one of the main claims of an already published scientific article. Its novelty therefore lies solely in the formalization of a claim, not the claim itself. The authors of such formalization papers consequently take credit for the way how the formalization is done, but not for the original claim (unless that claim happens to come from the same authors).

The content of a formalization paper is fully expressed in RDF in the form of nanopublications. Such a formalization paper can be shown in other formats to users, *e.g.*, in HTML or PDF, but these are just views of the same underlying RDF content. Our formalization papers consist of nanopublications in which the assertion contains the formalization of the scientific claim using the super-pattern (*Bucur et al., 2021*), the provenance points to the original paper of the claim, and the publication information attributes the author of the formalization. Figure 1 shows an example of such a nanopublication in the interface the participants of our study used to create them. The instantiated super-pattern in the assertion part refers to a context class, a subject class, a qualifier, a relation type, and an object class according to the super-pattern ontology (https://larahack.github.io/linkflows_superpattern/doc/sp/index-en.html). In the process of coming up with such a formalization, one often realizes that for some of the class slots of the super-pattern (*i.e.,* context, subject, and object class) the class that should be filled in to arrive at a correct formalization is not directly defined in any existing vocabulary or ontology and as such, this class might need to be minted as well. The provenance part of the nanopublication describes the "formalization activity" that was conducted in order arrive at this formalization from what is written in the source publication. The precise phrase from that source publication that was used can be quoted too.

## Tools

In order to publish formalization papers, class definitions, and all the other kinds of nanopublications (submissions, reviews, responses to reviews, and decisions), we use Nanobench (*Kuhn et al., 2021*) (https://github.com/peta-pico/nanobench). Figure 1 introduced above shows a screenshot of the publishing page of Nanobench. Publishing in Nanobench is based on templates, which are themselves expressed in nanopublications. The form shown in the screenshot is automatically generated based on the information found in several template nanopublications that we created and published for that purpose. All the application-specific behavior is therefore semantically represented in the templates, and Nanobench can flexibly be used for any other kind of data and workflow.

**Figure 1** Formalization paper template from Nanobench as used by the participants of our study.

The second tool that we are using, Tapas (*Lisena et al., 2019*) (https://github.com/peta-pico/tapas) is equally generic. It is a simple user interface component built on top of grlc (*Meroño-Peñuela & Hoekstra, 2016*) that allows to run template-based SPARQL queries on RDF triple stores. In our case, we run it on SPARQL endpoints provided by the nanopublication service network (*Kuhn et al., 2021*). We use Tapas to show aggregations and overviews of submissions and reviews. Figure 2 shows a screenshot of the main submission overview. Tapas by itself is read-only, but we connect to the Nanobench tool with links that lead to partially filled-in forms (*e.g.*, "click here to add review" in the screenshot).

## Field study design

In order to test our approach, we devised a field study where interested authors could submit formalization papers, which upon acceptance are to be published as a special issue in the journal Data Science (https://datasciencehub.net/) by IOS Press. The goal of this was to demonstrate for the first time that scientific articles can be formalized and therefore machine-interpretable including the main scientific claims. As a secondary goal, we wanted to find out whether nanopublications are a good technology for that, and whether it is feasible to represent also the entire submission and reviewing process within the same framework.

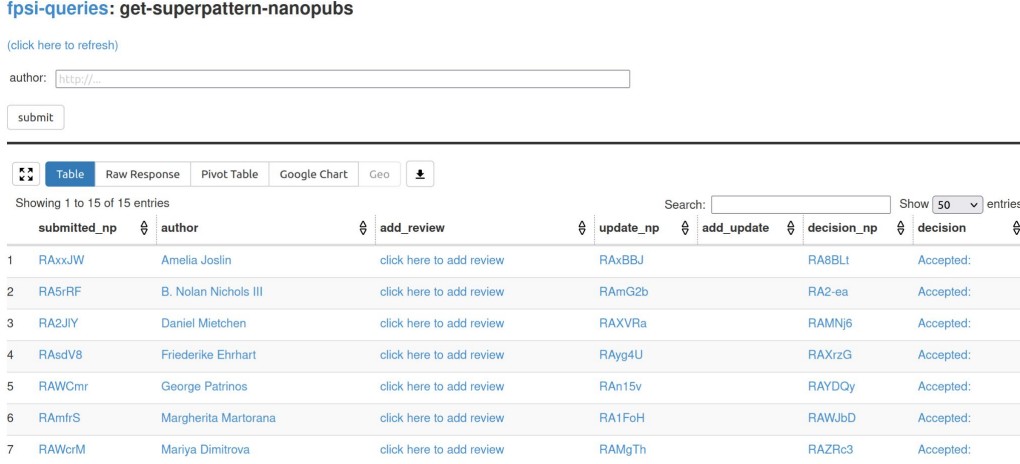

**Figure 2** **The Tapas interface listing submitted formalizations as the results of SPARQL queries over the nanopublication service network.**

Because the user interfaces we have at our disposal are still quite rough and technical, we restricted the set of possible authors and sent the call for papers on a by-invitation basis to selected groups of researchers who have previously worked or had experience with technologies like RDF and semantics. We expect to be able to build more accessible user interfaces in the future that can show the inherent complexity in a way that does not require technical skills, but how this can be achieved is out of scope for this work.

Participants to our field study, thus the authors of formalization papers, formalized their own previously published claim, or a claim from a paper published by others. In the latter case, the formalization paper authors take credit for the formalization of the claim but not for the claim itself. All submissions to this special issue were peer-reviewed (also as nanopublications) using our previously proposed reviewing ontology (*Bucur, Kuhn & Ceolin, 2019*). Upon acceptance, these formalization papers are to be published in a journal at IOS Press, thereby giving them the same bibliometric status as other scientific articles, which leads to regular indexing in scientific article databases, counting of citations, and so on.

The authors received close guidance on how to represent a claim of their choosing in RDF using the super-pattern and nanopublications, and on the various stages of the publication process. Authors took part in several information sessions and discussion meetings and were provided at each step with helper materials, videos, and even direct assistance if needed. In total, 24 such individual sessions were organized from May to December 2021.

In order to define a formalization, sometimes some of the class slots (*i.e.,* context, subject, and object slots) of the super-pattern should be filled in with classes that are not yet defined in any existing vocabulary or ontology. In this case the authors first had to define these themselves, and they could do that also with the Nanobench tool loading a template for class definition. (Alternatively, they could also mint a new class identifier by

other means, such as creating it on Wikidata.) The assertion of a nanopublication defining a new class may look for example as follows (link to full nanopublication):

```
sub:STX1B-mutation a owl:Class ;
  rdfs:subClassOf wd:Q42918 ;
  rdfs:label "STX1B mutation" ;
  skos:definition "mutation in STX1B" ;
  skos:relatedMatch wd:Q18048867 .
```

Here, "mutation" from Wikidata (Q42918) is declared as super-class of the newly minted class "STX1B mutation", and "STX1B" (Q18048867) is linked as a related class.

Then the authors can publish their formalization in the form of a nanopublication using Nanobench (see Fig. 1), and afterwards they needed to submit it to the special issue using another Nanobench template, leading to an assertion like (link to full nanopublication):

```
<http://purl.org/np/RAGo62Hb_Bx1klF4pn1q1Ty40860e3A7Sz4hr2vojZ2wA>
  pso:withStatus pso:submitted ;
  frbr:partOf fpsi:DataScienceSpecialIssue .
```

All submitted formalizations were subsequently reviewed. All authors were encouraged to review other submissions, and these reviews were semantic, open, and non-anonymous. These reviews were again done in nanopublications with the Nanobench tool. Such an example of a nanopublication assertion that contains a review modeled using the reviewing ontology can be seen below (link to full nanopublication):

```
sub:comment a lfr:ReviewComment , lfr:ContentComment , lfr:NeutralComment ,
    lfr:SuggestionComment ;
  lfr:hasCommentText "Maybe the use of a causal relation like \"contributes to\"
                        can also be used here." ;
  lfr:hasImpact "1" ;
  lfr:refersTo <http://purl.org/np/RAGo62Hb_Bx1klF4pn1q1Ty40860e3A7Sz4hr2vojZ2wA> ;
  lfr:refersToMentioningOf sp:hasRelation .
```

In such a structured review (see more details in our previous research (*Bucur et al., 2020*)), it is possible to specify various aspects that the review addresses including the aspect it comments on (syntax, style or content), the positivity/negativity of the review, the impact and the action that needs to be taken by the authors as the reviews see it (whether it is compulsory to be addressed, a suggestion or no action needs to be taken by the authors) and the importance of the point made by the review for the overall quality of the formalization. In the above example, the review comment makes a neutral point about the content of the given formalization with an importance of 1 out of 5, and is marked as a suggestion for the authors. The specific part of the formalization that this review targets is the *sp:hasRelation* field, as indicated by the *refersToMentioningOf* relation.

Subsequently, authors of the submissions could respond to the received review comments, again in nanopublications, and update their submissions based on these review comments. This is an example of a response to a review comment (link to full nanopublication):

```
sub:comment a , lfr:ResponseComment lfr:DisagreementComment ,
    lfr:PointNotAddressedComment ;
```

```
lfr:hasCommentText "I don't think the original publication shows a causal
                    relationship. It seems to me only a correlation is proven." ;
lfr:isResponseTo
  <http://purl.org/np/RAio--7IbPa3_ZSG3GspUsXeWP2ZwMIzy4Kzos0yZ7NIw> ;
lfr:refersTo <http://purl.org/np/RAeRSya2qIYymsBxiqOZP_oaQpHXUVXiydKvPCFM-7DDQ> .
```

This response registers the agreement with the point made by the reviewer (whether the author agrees totally, partially or not at all) and if that point was addressed, partially addressed or not addressed at all by the author. Moreover, a link to the respective review is given using the *isResponseTo* relation, while the updated version of the formalization is indicated using the *refersTo* relation. In our example, we see that the author does not agree with the point made by the reviewer and hence did not address the point raised by him, and also gives a textual motivation on why this is the case.

Finally, the authors updated their formalizations with the same template as depicted in Fig. 1. The full final formalization nanopublication of the same example is shown in Fig. 3. For all updated submissions then a decision was made by us as the special issue editors about their acceptance. This decision was also represented as a nanopublication that looked as follows (link to full nanopublication):

```
<http://purl.org/np/RAeRSya2qIYymsBxiqOZP_oaQpHXUVXiydKvPCFM-7DDQ>
  dct:description "All review comments were addressed and the formalization looks
                  good." ;
  pso:withStatus pso:accepted-for-publication ;
  frbr:partOf fpsi:DataScienceSpecialIssue .
```

All formalizations reached a satisfactory level of quality, as indicated by the reviews and the authors' responses, and we therefore accepted all 15 submissions for publication.

In order to show the accepted papers in the special issue as if they were classical papers, to integrate them in the publisher's content management system, and to make them connect to the existing bibliometric system, we semi-automatically created "classical views" in the form of HTML and PDF versions of the nanopublications, as can be seen in Fig. 4.

## User feedback

In order to evaluate the general idea of formalization papers, all participants to the field study were asked to give us their opinion and report on their experiences about the involved processes and concepts. This evaluation was performed by means of a structured questionnaire consisting of four main parts, each one evaluating different aspects of the workflow.

In the first part, we are interested in assessing the difficulty of conceptually understanding the formalization paper idea and the super-pattern, and of performing the formalization tasks. In part two, we focus on the difficulty of the technical aspects in the various submission, reviewing and revision stages. Part three addresses some more general aspects about the authors' experience and preferences. Authors were asked about their confidence in the formalization they published and about their interest of publishing such formalizations along their scientific publications in the future. We also asked them how important they think it is that all these steps are performed by the authors themselves (as

```
@prefix this: <http://purl.org/np/RAeRSya2qIYymsBxiqOZP_oaQpHXUVXiydKvPCFM-7DDQ> .
@prefix sub: <http://purl.org/np/RAeRSya2qIYymsBxiqOZP_oaQpHXUVXiydKvPCFM-7DDQ#> .
@prefix rdfs: <http://www.w3.org/2000/01/rdf-schema#> .
@prefix xsd: <http://www.w3.org/2001/XMLSchema#> .
@prefix dct: <http://purl.org/dc/terms/> .
@prefix np: <http://www.nanopub.org/nschema#> .
@prefix npx: <http://purl.org/nanopub/x/> .
@prefix nt: <https://w3id.org/np/o/ntemplate/> .
@prefix sp: <https://w3id.org/linkflows/superpattern/terms/> .
@prefix lfr: <https://w3id.org/linkflows/reviews/> .
@prefix wd: <http://www.wikidata.org/entity/> .
@prefix orcid: <https://orcid.org/> .
@prefix prov: <http://www.w3.org/ns/prov#> .

sub:Head {
  this: np:hasAssertion sub:assertion ;
    np:hasProvenance sub:provenance ;
    np:hasPublicationInfo sub:pubinfo ;
    a np:Nanopublication .
}

sub:assertion {
  sub:spi a sp:SuperPatternInstance ;
    rdfs:label "Mutations in STX1B are associated with epilepsy" ;
    sp:hasContextClass wd:Q5 ;
    sp:hasSubjectClass <http://purl.org/np/RAPVWYH0x-xyDa9PfBcGUFly3m1FNEO43KG9s0uH-y6yo#STX1B-mutation> ;
    sp:hasQualifier sp:frequentlyQualifier ;
    sp:hasRelation sp:cooccursWith ;
    sp:hasObjectClass wd:Q41571 .
}

sub:provenance {
  sub:assertion prov:wasGeneratedBy sub:activity .
  sub:activity a sp:FormalizationActivity ;
    prov:used <http://doi.org/10.1038/ng.3130> , sub:quote ;
    prov:wasAssociatedWith orcid:0000-0001-6501-0806 , orcid:0000-0002-6532-5880 , orcid:0000-0002-7979-9921 .
  sub:quote prov:value "Our results thus implicate STX1B and the presynaptic release machinery in fever-associated epilepsy syndromes" ;
    prov:wasQuotedFrom <http://doi.org/10.1038/ng.3130> .
}

sub:pubinfo {
  sub:sig npx:hasAlgorithm "RSA" ;
    npx:hasPublicKey "MIGfMA0GCSqGSIb3DQEBAQUAA4GNADCBiQKBgQCY36SLWPLee0SZGM108+7dyjGzKFYg9t09XuL3js13jO3CDzqAZygcrwbJsbLQMRHYvNf0Mkly1ePLgdb43NqEbXiDHC4o49nHjhi2bSWeRDJ4..." ;
    npx:hasSignature "QF+C9lXmczrn9cJNuimwLG45Mpntk2CcRIWbeNmKvfE9gmQ6MPKa/x6AfNgVQRnPNppJdDoWepK6m/+m8tWY1WQsXn0KZ8sER+graEHQYuE70Mz9JzuBTyYu0vpNj5jteoCve5fyvFkhkYVjoRK9..." ;
    npx:hasSignatureTarget this: .
  this: dct:created "2021-10-29T10:35:33.912+02:00"^^xsd:dateTime ;
    dct:creator orcid:0000-0001-6501-0806 ;
    npx:introduces sub:spi ;
    lfr:isUpdateOf <http://purl.org/np/RAGo62Hb_Bx1klF4pn1q1Ty40860e3A7Sz4hr2vojZ2wA> ;
    nt:wasCreatedFromProvenanceTemplate <http://purl.org/np/RAB_oy10D3XUP-zYlqGz7Uj58AsUXhEKeGqmRFg5LSgDM> ;
    nt:wasCreatedFromPubinfoTemplate <http://purl.org/np/RAA2MfqdBCzmz9yVWjKLXNbyfBNcwsMmOqcNUxkk1maIM> , <http://purl.org/np/RAOGu9Lh0BD4tbIRB9RG6RGRA_ObDh75NTbIqaWgxxs8M> ;
    nt:wasCreatedFromTemplate <http://purl.org/np/RAv68imZrEjfcp2rnEg1hzoBqEVc0cQMtp9_1Za08xNM4> .
}
```

**Figure 3**    A nanopublication view of a formalization paper.

## A formalization of one of the main claims of "Mutations in STX1B, encoding a presynaptic protein, cause fever-associated epilepsy syndromes" by Schubert et al. 2014[1]

Cite

**Article type:** Formalization Paper

**Authors:** Grouès, Valentin[a; *] | Moreno, Carlos Vega[b] | Satagopam, Venkata Pardhasaradhi[c]

**Affiliations:** [a] Luxembourg Centre for Systems Biomedicine, Université du Luxembourg, Luxemburg | [b] Luxembourg Centre for Systems Biomedicine, Université du Luxembourg, Luxemburg | [c] Luxembourg Centre for Systems Biomedicine, Université du Luxembourg, Luxemburg

**Correspondence:** [*] Corresponding author. E-mail: valentin.groues@uni.lu.

**Note:** [1] As RDF/nanopublication:
http://purl.org/np/RAeRSya2qIYymsBxiqOZP_oaQpHXUVXiydKvPCFM-7DDQ

**Keywords:** Human, STX1B mutation, epilepsy

**DOI:** 10.3233/DS-210051

**Journal:** Data Science, vol. Pre-press, no. Pre-press, pp. 1-3, 2022

**Received** 24 June 2021 | **Accepted** 17 November 2021 | **Published:** 28 February 2022

⬇ Get PDF   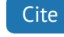

**Figure 4**    The human-readable view of a formalization paper, as it appears on the publisher's website https://content.iospress.com/articles/data-science/ds210051.

they did). Moreover, in this part, authors could give us their opinion with regard to the importance of having a ''classical view'' along with the nanopublication representation of their formalization paper. The fourth and final part of the questionnaire asked for the technical background of the authors. At the very end, the respondents could give further free-text feedback. The full questionnaire is available in our Supplemental Material.

## RESULTS

In this section we present the formalizations that resulted from our field study. We present a descriptive analysis of the generated data and analyze it also with the help of a network visualization. Finally, we report on the results from the user feedback questionnaire.

All the nanopublications that were created for all the submissions, formalization paper versions, the review comments, the responses to the review comments and the newly minted classes used in the formalizations together with the decisions are accessible online (https://github.com/LaraHack/formalization_papers_supplemental/tree/main/nanopubs), while the nanopublication index containing all these nanopublications has also been published (Nanopublication index: http://purl.org/np/RAkLJW7vIsnKKJDf1iswdgtFPQSo3lEG_z8DhHfD7dofE). Also, the final submissions for the special issue with formalization papers at the Data Science journal that was released in March 2022 (https://content.iospress.com/journals/data-science/5/1) can be found online (https://github.com/LaraHack/formalization_papers_supplemental/tree/main/accepted_submissions).

### Analysis of formalizations

In total, we had an initial number of 20 people that replied to our call for articles (https://github.com/LaraHack/formalization_papers_supplemental/tree/main/call_for_papers) from 12 different institutions from the United States of America, Germany, Luxembourg, Bulgaria, and The Netherlands from fields like biomedicine, bioinformatics, health sciences, ecology, data science, and computer science. After an initial information session, out of the 20 authors that responded to the call for papers, 18 decided to continue their participation. All these 18 authors that responded to the call for formalization papers managed in the end to publish (upon acceptance) their articles in a special issue at the Data Science journal.

We had a total of 15 formalization paper submissions, 13 with individual authors and 2 with joint authorship. Out of the total of 18 authors, two of these have both an individual submission and a joint-authorship one. The super-pattern instantiations of the final accepted formalization paper submissions can be seen in Table 1. Here, the classes used to instantiate the super-patterns that comprise the formalizations are given for each submission: the context, subject and object classes for each submission are listed, together with the qualifier and relations selected from the SuperPattern ontology (_Bucur et al., 2021_). Each instantiation of the super-pattern can be interpreted as follows: ''Every thing of type [SUBJECT] that is in the context of a thing of type [CONTEXT] [QUALIFIER] has a relation of type [RELATION] to a thing of type [OBJECT] that is in the same context.''.

**Table 1** **Instantiated super-patterns accepted for publication in formalization papers in the Data Science special issue.** Submissions marked with ◇ are formalizations in which authors extracted a scientific claim from their own previously published article; classes minted using Nanobench are marked with *, while newly minted Wikidata classes are marked with **.

| | Context (in the context of all …) | Subject (things of type …) | Qualifier | Relation | Object (to things of type…) |
|---|---|---|---|---|---|
| 1 | early human adipogenesis* | regulatory element within the first intron of FTO* | generally | affects | expression of genes IRX3 and IRX5* |
| 2 | human motor neuron (Q101404862) | TAR DNA binding protein (Q21133247) | can generally | contributes to | transcription of stmn2* |
| 3 ◇ | dejellied fertilizable stage VI Xenopus laevis oocyte** | strong static magnetic field** | generally | affects | cell cortex (Q5058180) |
| 4 ◇ | (no context class) | genes associated with CAKUT** | sometimes | is same as | targets of vitamin A** |
| 5 ◇ | patient undergoing PCI* | pharmacogenomics guided clopidogrel therapy* | generally | enables | cost-effective treatment* |
| 6 | human (Q5) | smoothened signaling pathway | mostly | affects | astrocyte development |
| 7 ◇ | biodiversity data (Q28946370) | license with non-commercial clause* | generally | inhibits | data reuse (Q58023280) |
| 8 ◇ | release of OpenBiodiv knowledge graph* | triple in OpenBiodiv knowledge graph* | generally | is same as | semantic triple extracted from biodiversity literature* |
| 9 | UNC13A (Q18036664) | TAR DNA binding protein (Q21133247) | generally | inhibits | inclusion of cryptic exon |
| 10 ◇ | data set (Q1172284) | adherence to the FAIR guiding principles* | can generally | enables | automated discovery* |
| 11 | human (Q5) | NGLY1 deficiency | always | is caused by | dysfunction of ERAD pathway* |
| 12 | social group (Q874405) | relative neocortex size* | never | affects | social group size* |
| 13 | ecm bound cancer cell* | glycocayx bulk* | generally | increases | integrin clustering* |
| 14 | human (Q5) | STX1B mutation* | frequently | co-occurs with | epilepsy (Q41571) |
| 15 ◇ | digital humanities research* | usage of Linked Data Scopes* | can generally | contributes to | transparency (Q535347) |

In the same Table 1 we can also take note of the distribution of qualifiers and relations that were used in the instantiated super-patterns of the accepted formalizations. As such, the most used qualifier is "generally" in almost 47% of cases (7 formalizations), while its modal counterpart, "can generally" is next in 20% of cases (3 formalizations). While all positive, non-modal qualifiers defined in the ontology seem to be used at least once (in at least one formalization), the only negative qualifier used was "never", in almost 7% of cases (1 formalization). The most used qualifiers are positive with about 73% (11 formalizations) and modal positive with 20% (3 formalizations), while the negative qualifiers seem to be less common with about 7% (1 formalization) and the modal negative qualifiers were never used. In terms of the relations used, we observe that relations that express causal relations are the majority with 80% (12 formalizations), then the next used is the equivalency relation with almost 13% (2 formalizations), then with the smaller ratio, the ones about the spatio-temporal relations with only almost 7% (1 formalization), while the relations making numerical comparisons (the "compares to" relations) were never used.

Looking at Table 1, we see that the super-pattern instances exhibit quite a broad variety of scientific fields (bioinformatics, biomedicine, pharmacology, data science, computer science) mostly linked to the life sciences. Seven out of the 15 submissions contain a formalization in which authors extracted a scientific claim from their own previously published article (submission number marked with ◇). Additionally, out of the total 44 classes used in the formalizations, 22 new classes were minted using Nanobench (marked with *), while four were newly minted Wikidata classes (marked with **). 13 already-existing classes were reused from Wikidata (their Wikidata identifier is specified next to the class name) and 4 classes were referenced from other ontologies.

## Analysis of nanopublications

In this field experiment we used nanopublications to embed, not only the formalizations created, but also the entire publication process that these formalizations underwent. As such, the entire formalization papers creation and publication process was thoroughly documented and published in a formal and machine-interpretable way, made possible by making use of nanopublications as "FAIR data containers". All the nanopublications pertaining to the special issue with formalization papers at the Data Science journal have been retrieved from the nanopublication network (https://monitor.petapico.org/) and made available online after serialization in *trig* files (https://github.com/LaraHack/formalization_papers_supplemental/tree/main/nanopubs).

Table 2 shows the statistics about the nanopublications created during our field study. It shows a total of 15 submissions with their 15 corresponding super-pattern definitions; the content of these submissions is the one summarized in detail in Table 1. There are 25 updated super-patterns, indicating that some of the submissions were updated more than once. 34 new classes were minted in nanopublications as class definitions, which were subsequently used in the formalizations. With regard to the reviews received and the author responses, class definitions received an average number of around 3 reviews per class (46 review comments in total), while the super-pattern definitions had

**Table 2** Nanopublications created during the field study of the special issue with formalization papers at the Data Science journal.

| Icon | Type | Average number Per submission | Total |
|------|------|------|------|
| S | Submissions | 1.00 | 15 |
| F | Super-pattern definitions | 1.00 | 15 |
| C | Class definitions | 2.27 | 34 |
| R | Reviews of super-patterns | 7.93 | 119 |
| R | Reviews of class definitions | 3.07 | 46 |
| A | Responses to super-pattern reviews | 6.67 | 100 |
| A | Responses to class definition reviews | 2.27 | 34 |
| U | Updated super-pattern definitions | 1.67 | 25 |
| D | Decisions | 1.00 | 15 |

almost 8 review comments on average (119 review comments in total). In terms of the responses given to these reviews, the average responses to class definitions was a little over 2 (34 review comments in total), while the average number of responses to the review comments for the super-pattern definitions was about 6.7 (100 review comments in total).

In Fig. 5 we can see a graphical representation of all the special issue nanopublications, where each node represents such a nanopublication and the arrows between the nodes show how the nanopublications are linked semantically with each other. The legend for the node types indicated by color and letter code can be found in Table 2. For every formalization paper, we see a first formalization (F) together with a submission nanopublication (S). Later updated versions (U) of formalizations also link back to the initial formalization. The initial submissions received review comments (R), to which authors then answered with response nanopublications (A). Additionally, some of the formalization papers used newly minted classes (C), which then also received review comments and responses. The final decision (D) points to the finally accepted updated formalization. The edges (*i.e.,* arrows) of the graph indicate when a nanopublication is referring to another one by using its identifier in the assertion. The edges shown in red are *superseding* relations, pointing from a new version of a nanopublication to its previous version. This is how nanopublications, being immutable, are dealing with representing new versions.

## User feedback analysis

The 18 authors and co-authors of the formalization papers were asked to fill in the user feedback questionnaire. It was important for this questionnaire to be fully and reliably anonymous, as the authors needed to be able to give their honest opinions. This meant that we had to send reminders without knowing who already filled it in. After several rounds of reminders, we ended up getting 19 responses, meaning that at least one of the authors submitted two responses. Due to the anonymous nature of this questionnaire,

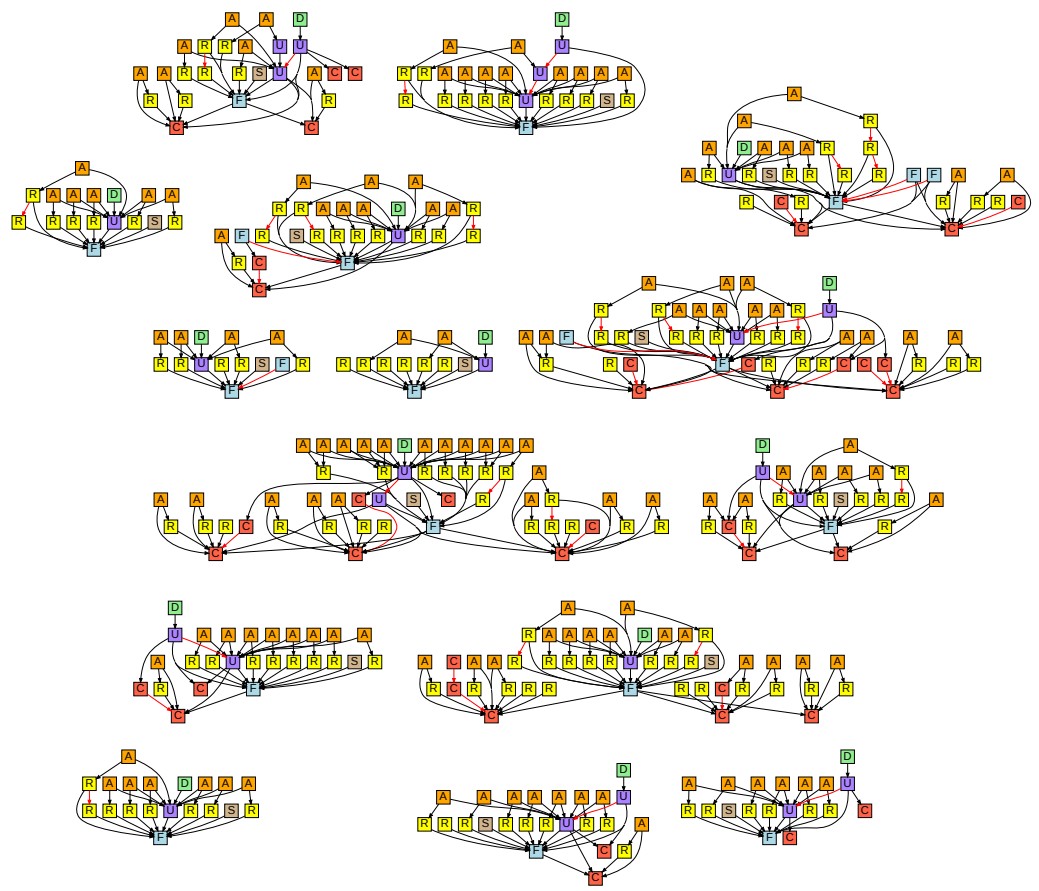

**Figure 5** **Graphical representation of all the submissions to the Data Science special issue with formalization articles (https://github.com/LaraHack/formalization_papers_supplemental/tree/main/visualization).** A click-able version with links to the nanopublications can be found online: https://raw.githubusercontent.com/LaraHack/fpsi_analytics/main/np-graph.svg.

it was not possible find out which responses were affected, and we have therefore to deal with such a dataset of slightly imperfect representation.

In Fig. 6 we see the results for the first part of the questionnaire. Authors expressed that it was rather easy to understand what a formalization paper is (with a score of 4.32 out of five). The elements of the super-pattern were found a bit harder to understand but still quite easy, with scores above 3.50. Finding an article from which to select a claim to formalize and to understand what the chosen claim really meant was also deemed easy, with scores of 3.74. The actual instantiation of the super-pattern with all its fields given the chosen claim was considered a little more difficult, with scores around 3.0, indicating medium difficulty roughly in the middle of *very difficult* and *very easy*. These results seem to suggest that the authors were able to understand the main formalization papers idea together with the super-pattern that comprises it, but when it came to the actual

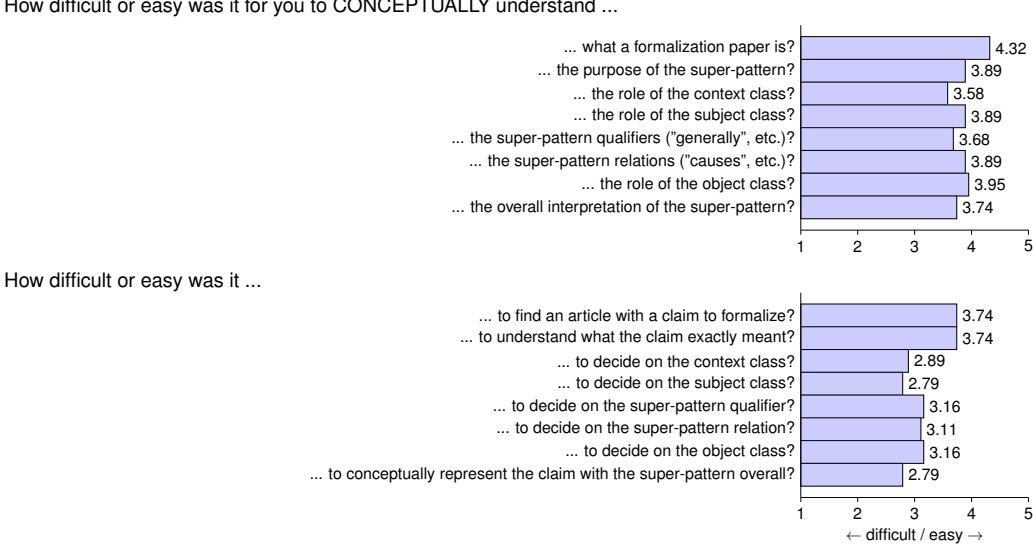

How difficult or easy was it for you to CONCEPTUALLY understand ...

How difficult or easy was it ...

← difficult / easy →

**Figure 6** Questionnaire Part 1: average answers from participants on conceptual aspects of formalization papers.

instantiation of the super-pattern (especially concerning the context and subject class), this was considered a little more difficult, but still on average far from very difficult.

In Fig. 7, we see the authors' responses with respect to technical difficulty. In terms of the tools used, we see that setting up and using Nanobench was considered easy enough (with a score of 3.30), while the Tapas interface seems a little harder to use (with a score of 2.76). The different tasks in the different stages all seemed to be between medium and easy on average, with the exception of the tasks to provide responses to reviews, which scored slightly below 3.0. The response nanopublications are indeed among the most complex ones, as they refer not only to the affected review but also to the updated formalization. Overall, while these results show room for improvement, they still seem favorable given that we were building upon generic and powerful tools without specific user interface design or polishing.

Figure 8 summarizes the assessment of more general aspects of formalization papers and also contains information about the authors' background. We see that authors have a high confidence in the quality of their formalization, with an average score of 4.0, and that they are interested in the future publication of such a formalization along their scientific publications, with a score of 4.05. The respondents very clearly stated that the classical view of formalization papers is important for website visitors, with a score of 4.68. Exposing the "naked" nanopublications to the website visitors with a nanopublication view was found to be much less important (3.32).

The authors indicated that they have, on average, a high level of knowledge on the topics of knowledge representation, knowledge graphs, Linked Data, and ontologies/vocabularies, with scores from 3.89 to 4.00. Their background in nanopublications, formal

# PeerJ Computer Science

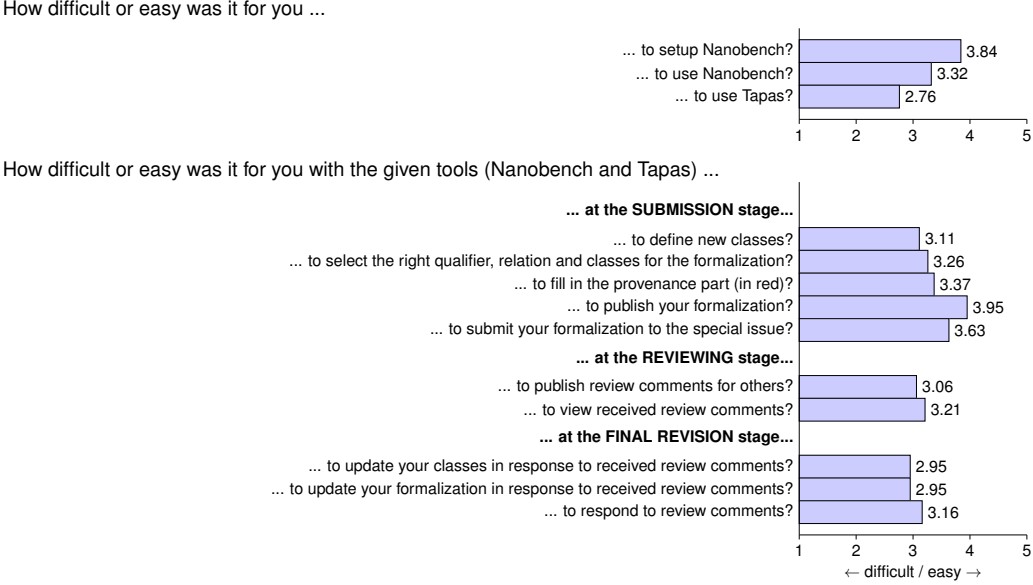

**Figure 7** Questionnaire Part 2: average answers from participants on technical aspects of formalization papers.

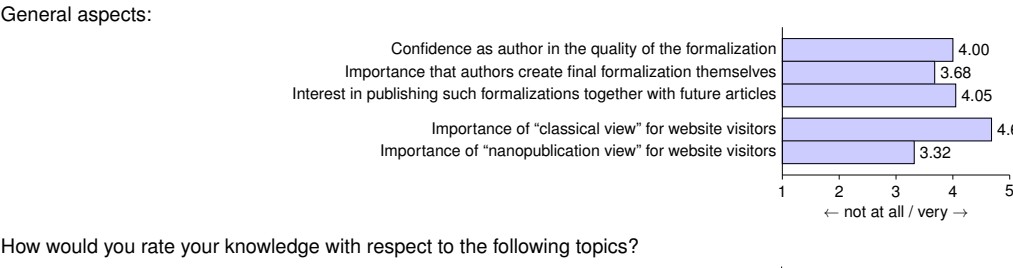

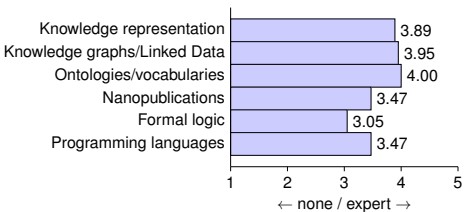

**Figure 8** Questionnaire Parts 3 and 4: average answers from participants on general aspects about formalization papers and average rating of participants with respect to their background knowledge on various topics.

logic, and programming languages was significantly lower, on average, but still relatively high, with scores between 3.05 and 3.47.

A total of 10 out of the 18 authors used the free text feedback of the questionnaire. Eight of these 10 respondents expressed their excitement about the field study and found the formalization paper concept and the whole publication process interesting and useful. However, half of these respondents also mentioned that the overall process proved to be a

little more difficult than they expected, due to the tools used maybe being too technical. One author also pointed out that multiple formalizations can be written for the same claim by choosing the context, subject and object classes differently and expressed the worry that this would decrease the interoperatibility or utility of formalizations especially when aggregating or mining them. This is a reasonable point to make, but due to the fully formal semantics, syntactic differences are in principle not hindering this kind of interoperability. Overall, the super-pattern, the formalization paper concept, and the nanopublication-based publication workflow seem to have been well-accepted and understood by the participants, and many of them showed an enthusiastic reaction.

## DISCUSSION AND CONCLUSION

The publication of the special issue with formalization papers at the Data Science journal shows not only that nanopublications and the super-pattern can be used to implement the basic steps and entities of a journal workflow, but also that authors of such formalization papers can be taught to use these in order to publish in a novel journal publication workflow as the publication of the special issue demonstrates. Our results show that the super-pattern can be well understood conceptually and despite the fact that from a practical standpoint applying it seems to be more difficult, its application remains perfectly feasible. Furthermore, we saw in our field study that even if the current general-purpose tools can be considered a viable solution, these are not necessarily easy to use, but they still remain a good tool for the purpose of publishing formalization papers. Moreover, considering the formalization papers, authors seem confident with regard to the quality of their publications and seem interested in publishing such formalizations in the future.

In future work, we plan to take the next logical step by publishing novel claims in this way from the start, and not depend on claims from already-published papers. These contributions will then also have to be accompanied by statements about the methods, equipment, and all other relevant scientific concepts, and can include not just the high-level claim but more lower-level ones, possibly all the way down to the raw data. This representation would then ideally cover the entire scientific workflow, starting from a motivation, leading to the design and execution of a study, and ending in new scientific insights. Such fully formalized scientific contributions can be seen as a major step—even a breakthrough—for the Semantic Web and Open Science movements and will bring us closer to a world where machines can interpret scientific knowledge and help us organize and understand it in a reliable and transparent manner.

## ACKNOWLEDGEMENTS

The authors would like to thank Stephanie Delbeque, Maarten Fröhlich and Johan Oomen for providing their insight and expertise.

### Funding

This work was funded by Vrije Universiteit Amsterdam, IOS Press and The Netherlands Institute for Sound and Vision. There was no additional external funding received for this study. The funders had no role in study design, data collection and analysis, decision to publish, or preparation of the manuscript.

### Grant Disclosures

The following grant information was disclosed by the authors:
Vrije Universiteit Amsterdam.
IOS Press.
The Netherlands Institute for Sound and Vision.

### Competing Interests

The authors declare there are no competing interests.

### Author Contributions

- Cristina-Iulia Bucur conceived and designed the experiments, performed the experiments, analyzed the data, performed the computation work, prepared figures and/or tables, authored or reviewed drafts of the article, organized meetings with participants, created helper materials and movies, gave presentations throughout the stages of the field study, and approved the final draft.
- Tobias Kuhn conceived and designed the experiments, performed the experiments, analyzed the data, performed the computation work, prepared figures and/or tables, authored or reviewed drafts of the article, and approved the final draft.
- Davide Ceolin conceived and designed the experiments, authored or reviewed drafts of the article, and approved the final draft.
- Jacco van Ossenbruggen conceived and designed the experiments, authored or reviewed drafts of the article, and approved the final draft.

### Data Availability

The call for papers, nanopublications for the special issue together with the accepted submission in "classical view", visualization and questionnaire are available in the Supplemental Files and at Github: https://github.com/LaraHack/formalization_papers_supplemental; LaraHack. (2022). LaraHack/formalization_papers_supplemental: formalization_papers_supplemental-v1.0 (v1.0). Zenodo. https://doi.org/10.5281/zenodo.6384300.

The queries to extract the nanopublications from the nanopublication decentralized network and the scripts for creating the nanopublication index and for the graphical visualization are available at Github: https://github.com/LaraHack/fpsi_analytics.

## Supplemental Information

Supplemental information for this article can be found online at http://dx.doi.org/10.7717/peerj-cs.1159#supplemental-information.

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
