# Peer review of "Nanopublication-based semantic publishing and reviewing: a field study with formalization papers"

_PeerJ Computer Science, doi:10.7717/peerj-cs.1159_

## Round 0.1 · original submission · Major Revisions

The reviewers have provided long and detailed reviews. They have noted fundamental issues that should be addressed, both to ensure that the manuscript has practical relevance and to ensure that the content is sufficient to describe and justify the approach. One reviewer asked you to cite specific articles. Although it may or may not be helpful to cite those specific articles, it highlights a potential need to be more comprehensive in the works cited.

Reviewer 1 ·

Basic reporting

The authors describe an effort to publish versions of original papers – or essential claims of original papers – in the nanopublication format, in RDF, using a “super-pattern” version of nanopublications structured as an ordered series of slots: Context class, subject class, qualifier, relation, object class. The implied but not rigorously stated claim is that publishing scientific articles natively as nanopublications using this pattern will make it more possible “for researchers in different disciplines to keep up-to-date with the recent findings in their field of study” (from the Abstract).
The language is clear, unambiguous, and professional, with no obvious grammar errors, and the structure is appropriate.

The authors are admirably persistent in continuing to work through the nanopublications concept, in which they have an established publications record. However, this paper does not state its research aim clearly or contextualize it well. the scientific question is neither well-formulated nor well-referenced. Yes there are a lot of references but not to the critical part of their argument.

The authors have much more to do to show that their approach has applicability to the problem they claim to address, and is not merely an exercise for a limited audience of computer scientists. Their purpose as they state it is to fill a gap in the existing ecosystem of biomedical publishing. They claim this can be done by changing the form in which articles are published and adding machine interpretability. They have to document this gap and show that the particular model they chose fills or has demonstrable promise to fill it; and that it is an important gap. They don’t do this:

To be very specific:
- lines 33-40 there are no references provided to show importance of the problem and any prior work on solutions; why prior solutions are inadequate; how and why this solution would solve the problem; in the Background section there is more referencing but the logical flow of the argument leaves a unreferenced gaps as follows:
- lines 113-114 Wong 2019 does show greater volumes of publications - but there is then a leap to "methods...to make scientific articles more machine-readable" as a solution without any evidence that this will be useful; it is handwavy; I feel certain that the authors could provide some beteer referencing and they should – otherwise the potential applicability of the proposed solution is left out - and it is quite important. They also ought to show what other work may have been done to handle the “too much information” problem, and how that problem is seen in comparison to others by the biomedical community.
- lines 121-122: "While the results can be very valuable there are also clear limitations, with the resulting data needing almost always manual curation to achieve decent quality"
- but this article itself is in essence a curation demonstration - they describe volunteers re-representing claims from several original articles as nanopublications – and the quality control is done here by questionnaire to the curators asking them whether they are satisfied with the results.
- lines 124-127 "A significant number of vocabularies and ontologies in many various domains have been developed, which are now ready to be used for scientific knowledge representation…However, they remain often difficult to find, access and understand due to the lack of documentation, versioning problems, and unresolvable URIs, among other things"
- it is hard to understand this statement which implies ontologies are not being used when in fact they are used intensively *by trained curators* working in biomedical knowledgebases such as UniProt.
- if in reality the authors of this article want the rich corpora of biomedical ontologies to be used by the primary producers of biomedical knowledge, this rather ignores that biomedical curation is a recognized discipline in knowledge production, requiring specialized training and having its own professional society.
- lines 229-231 "[when] the class that should be filled in to arrive at a correct formalization is not directly defined in any existing vocabulary or ontology ...this class might need to be minted as well" but now we have a tag cloud in which the tags are new un-curated “classes” not connected to existing ontologies – this is why biocuration is a specialized discipline and speaks to integrability.

The Figures could be improved both as to selection and content.
- Figure 3 seems marginally relevant if at all.
- Figure 4: the text is far too small to read without difficulty; the formalization of this paper's main claim "Mutations in STX1B are associated with epilepsy" loses fidelity from a straight rendering of the title "Mutations in STX1B, encoding a presynaptic protein, cause fever-associated epilepsy syndromes" or the last line of the abstract "Our results thus implicate STX1B and the presynaptic release machinery in fever-associated epilepsy syndromes" - note that the finding is about STX1B and fever-associated epilepsy syndromes, not simply epilepsy, and that STX1B is said to cause, not simply be associated with such syndromes; the formalization text does not have a link to STX1B in any ontology.
- Figure 5: Shows a "readable" view of a structured abstract of a non-biomedical paper (Wilkinson et al 2016), written by a data scientist. Is this one of the papers actually produced in this study? A structured abstract would be particularly easy to represent as a nanopublication, while the critical content areas of a real paper are not shown. As is well-known, abstracts often problematically do not quite conform to the contents of the actual paper. I was looking for some illustration of how the human-readable formulation of one of the annotated papers would appear.

Experimental design

As noted above, the research question is not well-posed or contextualized. The "experiment" or annotation exercise is not shown to be relevant to transforming the publication methods of the majority of biomedical researchers, but deals only with a small self-selected group who then self-evaluates. As a pilot study this would be far more convincing if actual biomedical researchers (no data scientists or computer scientists allowed) would be asked to evaluate the results.

The raw data and software do not seem to be provided.

Validity of the findings

As noted earlier, the evaluation was a questionaire to the people who produced the annotations, who were a self-selected group and should have been limited to biomedical researchers to keep the techno-people out. Biomedical researchers can be quite technical but the majority of lab workers I have known are quite allergic to the computer scientists' way of thinking and so this needs to be taken into account in both the production and the evaluation of the "formalization papers".

A point I would also make to the authors is that their example on lines 71-79 taken from Felix and Barrand 2002 is troubling in terms of loss of fidelity and missing the main point of the article. They say (line 79) that the claim can be rendered informally as

“whenever there is an instance of transient oxidative stress in the context of an instance of a rat brain endothelial cell, then generally (meaning in at least 90% of the cases), that instance of stress has the relation of affecting an instance of Pgp expression.”

But in fact, let’s look at the article’s title:

“P-glycoprotein expression in rat brain endothelial cells: evidence for regulation by transient oxidative stress”

The title say the article is presenting evidence that transient oxidative stress -regulates- PGP expression, not merely -affects- it. And there is no statement that “whenever …(meaning in at least 90% of the cases)” in the article anywhere. Lastly, the context is actually not a rat brain, but an artificial experiment in a lab, which makes the experimental conditions, the “evidence for regulation by transient oxidative stress” in the title a critical detail.

The PGP authors state in the last line of the abstract that “oxidative stress, by changing Pgp expression, may affect movement of Pgp substrates in and out of the brain.” This is the key take-home from the article.

And the key question a scientific reader will ask is, what is the evidence that this artificial experiment mirrors what really goes on in a rat brain? And whether this content is present affects the utility of the re-representation.

It would be good for the authors to address these points.

Additional comments

I am sorry to be so negative about this article. But I think the authors need to address many of the points shown above to be ready for publication and to be relevant to the problem they claim to be addressing.

Reviewer 2 ·

Basic reporting

• The major weakness is the small sample size which reduces the power of the study and increases the margin of error.
• The caption and the format of some tables could be improved, e.g., in table 5, the caption is not self-descriptive.
• You must cite and discuss related work and compare their results with yours. 
• Some examples where the language could be improved include lines 40, 71, 86, 138, 398, …
• related work section is missing. even though you mentioned some in the background section.
• You have to clearly mention the new contribution(s) compared to your previous work (i.e (Bucur et al., 2019) and (Bucur et al., 2020))
• I do not see the importance of having Fig 3. how the timeline for the special issue with formalization papers at the Data Science
• journal is relevant to the content?
• the structure of table 2 is not clear, e.g what do you mean by the value 2.27 in `per submission` for class definitions column
some broken links:
• Fig 6: https://raw.githubusercontent.com/LaraHack/fpsi analytics/main/np-graph.svg
• footnote on page 11: https://github.com/LaraHack/formalization
• papers supplemental/tree/main/nanopubs

Experimental design

The research question is not clearly defined and the knowledge gap being investigated should be identified, and positioned in the literature. I suggested several relevant studies below.

Validity of the findings

The Analysis of Nanopublications is not sufficient. the authors just reporting some statistics about the nanopublications they collected. Authors' interpretations of these figures are needed.

Additional comments

This work is a good step toward making scientific articles machine-interpretable by expressing scientific claims in these articles with formal semantics (i.e in RDF format). They use the concept of nanopublications for this goal. The whole submission process, including reviews, responses, and decisions, is considered as well.

I have a question for the authors? Who is responsible to use your super-pattern template to represent the meaning of the scientific claims in formal logic? I expect the answer to be "article authors". If so, could you grantees that the authors (i.e outside the CS community) are aware of writing scientific claims in formal logic? And what motivates them to do so?

suggested relevant studies:
• Said Fathalla, Sören Auer, & Christoph Lange (2020). Towards the semantic formalization of science. In SAC '20: The 35th ACM/SIGAPP Symposium on Applied Computing, online event, [Brno, Czech Republic], March 30 - April 3, 2020 (pp. 2057–2059). ACM.
• Sahar Vahdati, Said Fathalla, Sören Auer, Christoph Lange, & Maria-Esther Vidal (2019). Semantic Representation of Scientific Publications. In Digital Libraries for Open Knowledge - 23rd International Conference on Theory and Practice of Digital Libraries, TPDL 2019, Oslo, Norway, September 9-12, 2019, Proceedings (pp. 375–379). Springer.
• Said Fathalla, Sahar Vahdati, Sören Auer, & Christoph Lange (2017). Towards a Knowledge Graph Representing Research Findings by Semantifying Survey Articles. In Research and Advanced Technology for Digital Libraries - 21st International Conference on Theory and Practice of Digital Libraries, TPDL 2017, Thessaloniki, Greece, September 18-21, 2017, Proceedings (pp. 315–327). Springer.

---

## Round 0.2 · Minor Revisions

One of the reviewers provided a second review and suggested minor revisions to your manuscript. Please look at the reviewer's comments and provide a response to them.

Reviewer 1 ·

Basic reporting

Basic reporting is acceptable.

Experimental design

Acceptable as a proof of principle.

Validity of the findings

Interesting. This is not a statistically grounded study, it is a proof-of-principle, which makes sense in the context the authors describe.

Additional comments

I feel the article could benefit if the authors consider the following additional points. I don't feel these are actually "minor" points but they could be minor revisions.

1. "Semantics" is a feature of both natural language of statements in formal logic
Authors consistently use the term "semantic" or "semantics" to describe expressions in formal logic designed to be machine interpretable using semantic web approaches. This may seem a bit parochial to general CS readers coming from outside the semantic web culture. Authors might reflect that "semantics" is a strong feature of both natural and artificial / formal languages and that extremely rapid progress is being made in deep learning NLP approaches. Authors might improve this article by simply ensuring that they qualify "semantics" as "formal machine interpretable semantics"' or similar phrasing. Future progress in filling the "gap" mentioned by the authors may well come from combining semantic web and deep learning approaches.

2. "Conditional probability” vs. Likert scale
The authors provide what they call a "conditional probability" term in their formal nanopublication statements. But I get the distinct impression that this term does not reflect the actual numerical results in an article – how could such a generalization be possible in any event? – rather it seems to represent a subjective likelihood asserted by the authors, i.e. a value in a Likert scale. This would be a simple numerical substitute for the complex linguistic hedging typically found wrapped around the textual claims in any scientific publication. I recommend that this be made clear in the text.

---

## Author Rebuttal · Round 0.2

# Rebuttal letter for "Nanopublication-Based Semantic Publishing and Reviewing: A Field Study with Formalization Papers" article

**Note: Authors' responses are in purple.**

## Editor's Decision MAJOR REVISIONS

The reviewers have provided long and detailed reviews. They have noted fundamental issues that should be addressed, both to ensure that the manuscript has practical relevance and to ensure that the content is sufficient to describe and justify the approach. One reviewer asked you to cite specific articles. Although it may or may not be helpful to cite those specific articles, it highlights a potential need to be more comprehensive in the works cited.

We are happy to receive your response regarding our submission and the comments received from the reviewers. We have considered thoroughly all the comments made by the reviewers and made the necessary changes. As suggested, we made some changes to emphasize more the practical relevance and the justification of our approach. Moreover, we have revised our list of references and have also decided to add some that were indicated by one reviewer. Please find our detailed responses below.

[# PeerJ Staff Note: It is PeerJ policy that additional references suggested during the peer-review process should only be included if the authors are in agreement that they are relevant and useful #]

## Reviewer 1

**Basic reporting**
The authors describe an effort to publish versions of original papers – or essential claims of original papers – in the nanopublication format, in RDF, using a "super-pattern" version of nanopublications structured as an ordered series of slots: Context class, subject class, qualifier, relation, object class.

The implied but not rigorously stated claim is that publishing scientific articles natively as nanopublications using this pattern will make it more possible "for researchers in different disciplines to keep up-to-date with the recent findings in their field of study" (from the Abstract).
Following your comment, we clarified and toned down the statement in the Abstract by mentioning that we are just making a first step in the direction of solving the problem expressed there:
*"...propose a first step in this direction [processing scientific articles in an automated fashion…by making articles machine-interpretable] by setting out to demonstrate…"*

The language is clear, unambiguous, and professional, with no obvious grammar errors, and the structure is appropriate.

The authors are admirably persistent in continuing to work through the nanopublications concept, in which they have an established publications record. However, this paper does not state its research aim clearly or contextualize it well.

In order to state our research aim more clearly and to contextualize it better, we have rephrased and added more explanations about this in the following paragraphs (the second one is new), and we also added references to our previous work to show how we use our previous work in our current approach:

*"In the work to be presented below, the main research goal is to combine all the elements we have previously worked on --- namely semantic representation of reviews (Bucur et al., 2019), scientific works as a networks of nanopublications (Bucur et al., 2020), and representing the main claims with the super-pattern (Bucur et al., 2021) --- in order to implement genuine semantic publishing and putting it to the test in a field study.*

*Whereas our current scientific publishing process works with narrative-based, natural language texts in the form of scientific articles that are in the best case later made more machine-interpretable by means of semantic annotations and semantic interlinking, we propose a different approach, one that considers semantics from the beginning. Therefore, the main aim of this research is to make a first step in the direction of publishing with formal semantics from the start, showing that it is possible to semantically represent not only scientific claims, but also the entire scientific publishing process without going through other intermediary processing stages."*

the scientific question is neither well-formulated nor well-referenced.

We have put our research question better into focus and rephrased it to reflect our research aim better:

*"Can nanopublications and the super-pattern enable a new mode of scientific communication where authors publish their scientific findings with formal semantics from the start?"*

Yes there are a lot of references but not to the critical part of their argument.

We have revised the references and decided to add and remove some of them to improve our argumentation. As such, we have also added a separate subsection called "Genuine semantic publishing" in the "Background" section where we specifically discuss the previous work that is closest to our proposed approach.

In the "Introduction", we added in the first paragraph some missing references (Uddin2015, Bhargava2017, Westergaard2017, Westergaard2018, Shukkoor2022). In the "Background" section we removed Wong2009 as it did not focus on issues we particularly address in our research, and added Muller2018 to mention some of the research done in the field of biomedical literature curation together with Slater2014 that was previously mentioned. In the new "Genuine semantic publishing" subsection we added references about novel BEL developments (Muller2018, Hoyt2018, Zucker2021, Domingo-Fernandez2018, Shao2021, Hoyt2019 and Khatami2021), attempts at genuine semantic publishing for biodiversity data (Penev2012, Penev2018, Penev2019), together with projects that have a similar long-term vision as ours (Fathalla2020 and Vahdati2019).

The authors have much more to do to show that their approach has applicability to the problem they claim to address, and is not merely an exercise for a limited audience of computer scientists. Their purpose as they state it is to fill a gap in the existing ecosystem of biomedical publishing.

Many of the participants were in fact not computer scientists, as we can see from the table below where we listed all authors who published in the special issue (as we also mention briefly in section 4.1, first paragraph).

Furthermore, the purpose of our research was not to fill a gap just in the biomedical publishing field, but to fill a gap in the publishing field as a whole. Considering the innovative and novel nature of this approach, however, we needed to make smaller practical steps towards our vision, which is why we made a call for formalization papers only to certain groups of researchers that had some previous experience working with ontologies. In future work, more user-friendly tools will be needed to make this idea more broadly available, so having such specialized knowledge will not be necessary anymore. As our approach is really a major paradigm shift in how we should publish in the future, it is in our view not feasible to implement and evaluate this at once, but small steps with practical concessions, such as the one we describe, are the only feasible option. We clarified these points also in the paper.

| No. | Name author | Main expertise of participant |
| --- | --- | --- |
| 1 | Amelia Joslin | data scientist |
| 2 | Nolan Nichols | bioinformatician |
| 3 | Daniel Mietchen | biophysicist |
| 4 | Friederike Ehrhart | biologist, bioinformatician |
| 5 | Chris Evelo | biologist, bioinformatician |
| 6 | George Patrinos | biologist |
| 7 | Margherita Martorana | computer scientist, bioinformatician |
| 8 | Mariya Dimitrova | bioinformatician |
| 9 | Lyubomir Penev | biologist and ecologist |
| 10 | Matthew Brauer | computer scientist |
| 11 | Michel Dumontier | biochemist, data scientist |
| 12 | Núria Queralt Rosinach | computer scientist, bioinformatician |
| 13 | Ricardo Usbeck | computer scientist |
| 14 | Russell Bainer | computational biologist |

| 15 | Valentin GROUÈS | computer scientist, bioinformatician |
|----|-----------------|-------------------------------------|
| 16 | Venkata Satagopam | bioinformatician |
| 17 | Carlos Vega Moreno | computer scientist, bioinformatician |
| 18 | Victor de Boer | computer scientist |

They claim this can be done by changing the form in which articles are published and adding machine interpretability. They have to document this gap and show that the particular model they chose fills or has demonstrable promise to fill it; and that it is an important gap.

We have added a new subsection "Genuine semantic publishing" in the "Background" section to show the gap that our research is trying to fill and to contextualize better our research, together with some other clarifications in the Introduction.

To be very specific:
- lines 33-40 there are no references provided to show importance of the problem and any prior work on solutions; why prior solutions are inadequate; how and why this solution would solve the problem;

We added Uddin2015, Bhargava2017, Westergaard2017, Westergaard2018, Shukkoor2022, but many others can be added as well.

in the Background section there is more referencing but the logical flow of the argument leaves a unreferenced gaps as follows:

- lines 113-114 Wong 2019 does show greater volumes of publications - but there is then a leap to "methods...to make scientific articles more machine-readable" as a solution without any evidence that this will be useful; it is handwavy; I feel certain that the authors could provide some beteer referencing and they should – otherwise the potential applicability of the proposed solution is left out - and it is quite important. They also ought to show what other work may have been done to handle the "too much information" problem, and how that problem is seen in comparison to others by the biomedical community.

We improved the logical structure of this part. We therefore removed the Wong 2019 reference and the phrase mentioning the "too much information" problem as this is indeed not central to our approach.

- lines 121-122: "While the results can be very valuable there are also clear limitations, with the resulting data needing almost always manual curation to achieve decent quality"
- but this article itself is in essence a curation demonstration - they describe volunteers re-representing claims from several original articles as nanopublications

Of course we agree that biomedical curation is an immensely useful and successful discipline and we did not want our article to imply anything to the contrary, but the main goal of our research was not curation. Our approach is to represent scientific claims with formal semantics from the start, so this kind of curation is no longer needed. But, in order to do this, we had to

make some practical compromises for our evaluation. Therefore, we started with existing claims from already published scientific articles, hence the findings were not original, but the approach, the processes, and the tools were designed so they would work in the same way for original findings. So, while our evaluation might have some similarity to curation (again, due to practical concessions we had to make), our approach does not. We clarified this also in the article.

– and the quality control is done here by questionnaire to the curators asking them whether they are satisfied with the results.
The quality control of the formalizations is not done with the help of the questionnaire alone. A main part of the quality control is done by analyzing the data from the full reviewing and decision process, as described in Sections 4.1 and 4.2. The questionnaire then adds to these results by showing that the authors were satisfied with the process and its outcomes.

- lines 124-127 "A significant number of vocabularies and ontologies in many various domains have been developed, which are now ready to be used for scientific knowledge representation…However, they remain often difficult to find, access and understand due to the lack of documentation, versioning problems, and unresolvable URIs, among other things"
- it is hard to understand this statement which implies ontologies are not being used when in fact they are used intensively *by trained curators* working in biomedical knowledgebases such as UniProt.
Indeed, vocabularies and ontologies are used extensively, especially by the biomedical community. We did not want to imply that they are not, but indeed, our emphasis was too much on the issues that come with their usage. We have therefore rephrased the above statement:
*"But, even though practical problems like finding, accessing and versioning among other things have been reported, these vocabularies and ontologies have proven to be extremely useful for example for biomedical literature curation (Slater, 2014; Muller et al., 2018)."*

- if in reality the authors of this article want the rich corpora of biomedical ontologies to be used by the primary producers of biomedical knowledge, this rather ignores that biomedical curation is a recognized discipline in knowledge production, requiring specialized training and having its own professional society.
As we also mentioned above, we absolutely agree that biomedical curation is a very successful discipline being very useful for the producers of biomedical knowledge, and one that requires special training and expertise. We do not want to suggest anything to the contrary, but, as our main research goal in this article was not curation, we did not touch upon this in detail. We adjusted the text of our paper to better get this across.

- lines 229-231 "[when] the class that should be filled in to arrive at a correct formalization is not directly defined in any existing vocabulary or ontology ...this class might need to be minted as well" but now we have a tag cloud in which the tags are new un-curated "classes" not connected to existing ontologies – this is why biocuration is a specialized discipline and speaks to integrability.
In Table 1 we see the full specification of the classes used in the formalizations. While it is true that 22 out of the total of 44 classes (see the more detailed description in lines 389-396), were

newly minted classes in the Nanobench tool, all these new classes are linked to WikiData by means of subclasses and related classes. For example, if we look (click on it to see the nanopublication) at the context class "ecm bound cancer cell" (in Table 1, formalization 13), this class is defined in the class definition nanopublication as a subclass of the "cancer cell" class in Wikidata (https://www.wikidata.org/wiki/Q4118072) and is related to the "extracellular matrix" class in Wikidata (https://www.wikidata.org/wiki/Q193825). So, all of the newly minted classes are semantically linked to WikiData, and a few even have more complex OWL definitions, like the "usage of Linked Data Scopes" subject class (in Table 1, formalization 15), which uses the restriction owl:someValuesFrom on the "uses" WikiData property (https://www.wikidata.org/wiki/Property:P2283).

The definitions of the classes themselves were also "curated" by the authors not only with the help of the editors of this special issue, but they were also mostly "curated" and improved with the help of the reviews received for the newly minted classes. As such, not just the formalizations themselves received reviews, but also the newly minted classes. Moreover, as these classes have a unique URL, they can also be linked later to other ontologies or classes in other ontologies.

The Figures could be improved both as to selection and content.
- Figure 3 seems marginally relevant if at all.
We agree and have therefore removed Figure 3.

- Figure 4: the text is far too small to read without difficulty;
We think the text size is fine when printed on A4 paper and if it is read on a computer, readers can even zoom in on the image in case they want to see the text a little bigger. Therefore we left the font size as it was, but if the reviewer insists, we will adjust the text in the nanopublication to make it bigger.

the formalization of this paper's main claim "Mutations in STX1B are associated with epilepsy" loses fidelity from a straight rendering of the title "Mutations in STX1B, encoding a presynaptic protein, cause fever-associated epilepsy syndromes" or the last line of the abstract "Our results thus implicate STX1B and the presynaptic release machinery in fever-associated epilepsy syndromes" - note that the finding is about STX1B and fever-associated epilepsy syndromes, not simply epilepsy, and that STX1B is said to cause, not simply be associated with such syndromes;
While we agree to some of the comments and concerns mentioned by the reviewer for submission 14 from Table 1, we provide the following clarifications:
1) While it is true that such formalizations might not be perfect, this shows the importance of having the original authors of the scientific articles from which the claims were taken make the formalization of their own claims as they have all the expertise to understand them best.
2) This formalization of the claim that makes use of the super-pattern is a bit less specific than the original claim, but it is still a correct formalization.

3) Indeed, in this particular article from which the example for the formalization was taken, there is a slight difference on how the relation in question is represented in the title as compared to the abstract. While in the title the causality is directly specified, in the abstract, the causality is not explicitly specified. Possibly, the formalization paper authors, who are experts in the field, chose to focus on the abstract as their main source of input. This shows again the importance that the authors of the claims themselves make the formalization of their claim. This just shows us that we need more precise ways of communicating even smaller bits of knowledge like claims and findings and our approach can help us detect, discuss and correct such errors, which is an advantage in itself.

the formalization text does not have a link to STX1B in any ontology.
As we also explain above, all the classes that were created were linked to an existing Wikidata class (or other ontology). As such, "STX1B mutation" (see Table 1, formalization 14) is defined in the class definition nanopublication as a subclass of the "Mutation" class in Wikidata (https://www.wikidata.org/wiki/Q42918) and is related to the "STX1B" class in Wikidata (https://www.wikidata.org/wiki/Q18048867).

- Figure 5: Shows a "readable" view of a structured abstract of a non-biomedical paper (Wilkinson et al 2016), written by a data scientist. Is this one of the papers actually produced in this study?
We changed Figure 5 to represent the same example as in the previous figures, showing the appearance of the final human-readable view of the published formalization paper of the special issue.

A structured abstract would be particularly easy to represent as a nanopublication, while the critical content areas of a real paper are not shown. As is well-known, abstracts often problematically do not quite conform to the contents of the actual paper. I was looking for some illustration of how the human-readable formulation of one of the annotated papers would appear.
The formalization paper and the way it is represented in a human-readable form can be seen in the published special issue here (https://content.iospress.com/journals/data-science/5/1 ). The specific human-readable form of the "formalization paper" represented in Figure 5 can be seen here: https://content.iospress.com/articles/data-science/ds210051. These links were also added in the new version of our paper.

**Experimental design**
As noted above, the research question is not well-posed or contextualized.
To clarify this, we added a phrase in the Introduction (see above), revised the references and also added a separate subsection titled "Genuine semantic publishing" in the "Background" section. We also rephrased the research question.

The "experiment" or annotation exercise is not shown to be relevant to transforming the publication methods of the majority of biomedical researchers, but deals only with a small

self-selected group who then self-evaluates. As a pilot study this would be far more convincing if actual biomedical researchers (no data scientists or computer scientists allowed) would be asked to evaluate the results.

As we also mentioned above, our approach is not just for biomedical researchers, but for researchers as a whole, despite the fact that we performed our field experiment only with certain specialized researchers (with some computer science experience as well). It was the first time such an experiment was performed and the tools used for publication were also not user-friendly in the sense of them still needing some development with regard to the interfaces used. Nevertheless, our field experiment shows how well our approach works in a real setting (as participants do not actually self-evaluate, but evaluate their experience of the process of publishing a formalization). The questionnaire is just a part of the evaluation and the formalizations the participants created were reviewed by other researchers.

With this field experiment we wanted to show that a new way of publishing is possible and we have made the first steps in this direction. As we wrote above, many of the participants were not computer scientists. Moreover, 7 out of 15 submissions formalized their own results, so they were the prime experts on the subject. See also our more elaborate answer to the same points above.

The raw data and software do not seem to be provided.

All materials are here: https://github.com/LaraHack/formalization_papers_supplemental. The Nanopub software link and the Tapas interfaces are provided as well.

**Validity of the findings**
As noted earlier, the evaluation was a questionaire to the people who produced the annotations, who were a self-selected group and should have been limited to biomedical researchers to keep the techno-people out. Biomedical researchers can be quite technical but the majority of lab workers I have known are quite allergic to the computer scientists' way of thinking and so this needs to be taken into account in both the production and the evaluation of the "formalization papers".

We agree that the tools and the user interfaces need to be further developed based on feedback from the communities to make them polished and accessible for broader future use, but for the reasons stated above, we believe this is an important and valid first step.

A point I would also make to the authors is that their example on lines 71-79 taken from Felix and Barrand 2002 is troubling in terms of loss of fidelity and missing the main point of the article. They say (line 79) that the claim can be rendered informally as

"whenever there is an instance of transient oxidative stress in the context of an instance of a rat brain endothelial cell, then generally (meaning in at least 90% of the cases), that instance of stress has the relation of affecting an instance of Pgp expression."

But in fact, let's look at the article's title:

"P-glycoprotein expression in rat brain endothelial cells: evidence for regulation by transient oxidative stress"

The title say the article is presenting evidence that transient oxidative stress -regulates- PGP expression, not merely -affects- it. And there is no statement that "whenever …(meaning in at least 90% of the cases)" in the article anywhere. Lastly, the context is actually not a rat brain, but an artificial experiment in a lab, which makes the experimental conditions, the "evidence for regulation by transient oxidative stress" in the title a critical detail.

The PGP authors state in the last line of the abstract that "oxidative stress, by changing Pgp expression, may affect movement of Pgp substrates in and out of the brain." This is the key take-home from the article.

And the key question a scientific reader will ask is, what is the evidence that this artificial experiment mirrors what really goes on in a rat brain? And whether this content is present affects the utility of the re-representation.

It would be good for the authors to address these points.

These are valid observations and we would like to respond by also referring to the points above made about submission 14 from Table 1. As we mention there as well, formalizations can be correct and useful even when they are not as specific as they could be, and this just shows the importance of authors themselves making the formalizations.

Moreover, our formalizations are supposed to cover the core meaning of the claim, and not necessarily all the syntactic components that the original sentence used to convey this meaning. It is true that "in at least 90% of the cases" (the "generally" qualifier) is implicit in the existing text, but it is important for a full formalization. We need the strength of the relation to be expressed explicitly in order to make it fully machine-interpretable. We extended our discussion of the given example in the Introduction to clarify these points to the reader. At the same time, because authors are not forced to be explicit about this, we need to make a best guess for such formalization papers, but according to our approach, the researchers themselves would specify that piece of information in the future.

Also, the use of the causal relation "affects" as it is defined in the SuperPattern ontology in this case seems an appropriate choice of a relation, even though not as precise as it could be. A more precise representation would have been possible, if modeled differently, e.g. "causes [regulation of pgp expression]", but the given statement is still reasonably precise in our view. We have shown in our previous research (Bucur2021), that there are often several correct formalizations for a single claim, and knowledge representation experts can agree on the best formalization in the vast majority of the cases. We also showed that the super-pattern can be applied in a fairly consistent and reliable manner, despite the inherent complexity and difficulty of such a task.

The distinction between the experiment setting and the setting in which the claim is supposed to hold is indeed a very important one. In our view, a lab experiment like the one described in Felix2002 is just a piece of evidence (weak or strong) for an effect that is thought to happen in the real world (so in rat brains in this case). Nanopublications are in fact designed to record exactly such kind of information with their provenance in a clean and unambiguous way. The assertion of the nanopublication contains in this case the effect in the real world (so about rat

brains) whereas the provenance part links this statement (which might be true of false) to the piece of evidence (in this case a lab experiment) that made the author of the nanopublication state it. (For formalization papers, the piece of evidence is in fact not an experiment but a paper where this experiment is described. Therefore the provenance of formalization papers points to a paper and not to an experiment.)

**Additional comments**

I am sorry to be so negative about this article. But I think the authors need to address many of the points shown above to be ready for publication and to be relevant to the problem they claim to be addressing.

We thank you for your feedback and your comments, we hope we have managed to address and respond in a satisfactory manner to all the points you have raised.

# Reviewer 2

**Basic reporting**

• The major weakness is the small sample size which reduces the power of the study and increases the margin of error.

The sample size is 18 authors and 15 published formalization papers. Considering that this was a first initiative of this kind and all the participants had to be guided throughout the process (with meetings, demos, instructions), we believe this to be a reasonable number for such a first step into an entirely new approach. Moreover the special issues for Elsevier journals in Social Science and Medicine, for example, generally comprise of 15-20 articles (https://www.journals.elsevier.com/social-science-and-medicine/policies/ssm-guidelines-for-special-issues) and for the other special issues for the Data Science journal at IOS Press (where the special issue was published) this number is in accordance with the previous issues where about 10-15 articles were published per issue. With our field study experiment we wanted to see if the approach we proposed can actually work in a real-life setting and we wanted to see how this can be done. Therefore, our main results are qualitative in nature in order to show that this is possible. We also argue that a larger number of participants in this study, where our focus was on the qualitative results, would have been harder to manage and beyond the point of what we wanted to prove.

• The caption and the format of some tables could be improved, e.g., in table 5, the caption is not self-descriptive.

We assume the reviewer meant the caption of Figure 5 (we don't have a Table 5). As such, we have rephrased the caption to make it more self-descriptive to: *"The human-readable view of a formalization paper, as it appears on the publisher's website".* Please let us know if you believe other changes would be needed.

Moreover, we checked all captions and made them self-descriptive where appropriate (Figures 4, 7, 8 and 9).

• You must cite and discuss related work and compare their results with yours.

Thank you for your suggestion. We have better contextualized our research in the "Introduction" to better clarify our research aim and also, after revising some of our references, we added a new subsection "Genuine semantic publishing" in the "Background" section. Please see the complete details of all the references added and removed from the different sections of the article in one of the responses we gave above. We copy it here for easier access:

In the "Introduction", we added in the first paragraph some missing references (Uddin2015, Bhargava2017, Westergaard2017, Westergaard2018, Shukkoor2022). In the "Background" section we removed Wong2009 as it did not focus on issues we particularly address in our research, and added Muller2018 to mention some of the research done in the field of biomedical literature curation together with Slater2014 that was previously mentioned. In the new "Genuine semantic publishing" subsection we added references about novel BEL developments

(Muller2018, Hoyt2018, Zucker2021, Domingo-Fernandez2018, Shao2021, Hoyt2019 and Khatami2021), attempts at genuine semantic publishing for biodiversity data (Penev2012, Penev2018 and Penev2019), together with projects that have a similar long-term vision as ours (Fathalla2020 and Vahdati2019).

• Some examples where the language could be improved include lines 40, 71, 86, 138, 398, …
Thank you for pointing these out. We checked and improved these and similar passages.

• related work section is missing. even though you mentioned some in the background section.
We cover our related work in the "Background" section. We have also added a new subsection there titled "Genuine semantic publishing" where we mention and list some references that put our research better into context. Moreover, we have added more relevant references and removed some. See the list of these above.

• You have to clearly mention the new contribution(s) compared to your previous work (i.e (Bucur et al., 2019) and (Bucur et al., 2020));
We better contextualized our research by adding a new phrase in the Introduction to better clarify our research aim and we also added references to our previous work to show how we use it in our current approach. Please see our response above, where we describe in detail these changes.

• I do not see the importance of having Fig 3. how the timeline for the special issue with formalization papers at the Data Science;
We followed this suggestion and removed the figure.

• journal is relevant to the content?
Yes, we believe this to be the case as it deals with formal semantics, knowledge representation, user interfaces and digital libraries, topics that can be found among the ones listed.

• the structure of table 2 is not clear, e.g what do you mean by the value 2.27 in per⊂mission for class definitions column
We have changed the title column *"per submission"* to *"average number per submission"* to better reflect the numbers in that column. As such, the 2.27 value means that, on average, authors have given a little over 2 responses to the reviews received for their class definitions. We hope this is clearer, if not we are open to further suggestions.

some broken links:
• Fig 6: https://raw.githubusercontent.com/LaraHack/fpsi analytics/main/np-graph.svg
• footnote on page 11: https://github.com/LaraHack/formalization
• papers supplemental/tree/main/nanopubs
These broken links were due to the automated process of generating the PDF from the Latex file we submitted on the PeerJ website; they should be fixed.

**Experimental design**

The research question is not clearly defined and the knowledge gap being investigated should be identified, and positioned in the literature. I suggested several relevant studies below.

Please see our detailed responses above. We have made changes to better contextualize our research and emphasize better the gap we are investigating, while also slightly changing our research question to reflect this. Moreover, we have added two of the three relevant studies suggested below.

**Validity of the findings**

The Analysis of Nanopublications is not sufficient. the authors just reporting some statistics about the nanopublications they collected. Authors' interpretations of these figures are needed.

In section 4.1 we provide an analysis of the formalizations submitted to the special issue, while in section 4.2. we describe the nanopublications that embed, not only the formalizations, but also the entire publication process (with reviewing, responding to those reviews, submitting updated formalizations, the final publication decision). We have added a new paragraph describing in detail the qualifiers and relations used in the formalizations in section 4.1 and also added a new paragraph in the beginning of section 4.2, while adding some more detailed numbers in the text where Table 2 is described.

If you would be so kind as to provide us with more suggestions on what more we should add, we would be happy to provide and add more information.

**Additional comments**

This work is a good step toward making scientific articles machine-interpretable by expressing scientific claims in these articles with formal semantics (i.e in RDF format). They use the concept of nanopublications for this goal. The whole submission process, including reviews, responses, and decisions, is considered as well.

I have a question for the authors? Who is responsible to use your super-pattern template to represent the meaning of the scientific claims in formal logic? I expect the answer to be "article authors". If so, could you grantees that the authors (i.e outside the CS community) are aware of writing scientific claims in formal logic? And what motivates them to do so?

Ideally, the authors of the original scientific claim themselves would use the super-pattern template to represent the meaning of the scientific claim they wrote about.

While this needs further research, we assume that user-friendly tools will become available in the future that will allow researchers to make such representations in a quick and easy-to-understand manner, which won't require them to be aware of the full logical representation and theory behind the super-pattern. Once the writing of claims in formal logic would be adapted, automatic aggregations and visualizations of the latest state of the art research would be possible, together with a higher visibility and easy way to connect scientific claims from various research experiments and groups, enabling easy searches, findings and collaborations between the research community. And we believe that these can motivate further researchers for using such an approach like the one we propose.

suggested relevant studies:

• Said Fathalla, Sören Auer, & Christoph Lange (2020). Towards the semantic formalization of science. In SAC '20: The 35th ACM/SIGAPP Symposium on Applied Computing, online event, [Brno, Czech Republic], March 30 - April 3, 2020 (pp. 2057–2059). ACM.

• Sahar Vahdati, Said Fathalla, Sören Auer, Christoph Lange, & Maria-Esther Vidal (2019). Semantic Representation of Scientific Publications. In Digital Libraries for Open Knowledge - 23rd International Conference on Theory and Practice of Digital Libraries, TPDL 2019, Oslo, Norway, September 9-12, 2019, Proceedings (pp. 375–379). Springer.

• Said Fathalla, Sahar Vahdati, Sören Auer, & Christoph Lange (2017). Towards a Knowledge Graph Representing Research Findings by Semantifying Survey Articles. In Research and Advanced Technology for Digital Libraries - 21st International Conference on Theory and Practice of Digital Libraries, TPDL 2017, Thessaloniki, Greece, September 18-21, 2017, Proceedings (pp. 315–327). Springer.

Thank you for your suggestions. These are very interesting and indeed very relevant. We included two of these references: Fathalla2020 and Vahdati2019.

---

## Round 0.3 · accepted · Accept

Thank you for addressing the reviewers' comments.